# Efficacy and Safety of Vaccinations in Geriatric Patients: A Literature Review

**DOI:** 10.3390/vaccines11091412

**Published:** 2023-08-24

**Authors:** Tiziana Ciarambino, Pietro Crispino, Pietro Buono, Vincenzo Giordano, Ugo Trama, Vincenzo Iodice, Laura Leoncini, Mauro Giordano

**Affiliations:** 1Internal Medicine Department, Hospital of Marcianise, ASL Caserta, 81031 Caserta, Italy; 2Direzione di Staff Direzione Generale Tutela per la Salute Regione Campania, 80143 Naples, Italy; p.buo@libero.it (P.B.); u.tra@gmail.com (U.T.); 3Internal Medicine Department, Hospital of Latina, ASL Latina, 04100 Latina, Italy; p.cri@libero.it; 4UOD 02 Prevenzione Regione Campania, 80143 Naples, Italy; 5ASL Caserta, Direttore Sanitario Aziendale, 81100 Caserta, Italy; 6ASL Caserta, Direttore Sanitario, P.O. Marcianise, 81025 Marcianise, Italy; 7Department of Advanced Medical and Surgical Science, University of Campania, L. Vanvitelli, 81100 Naples, Italy; m.gior@libero.it

**Keywords:** elderly, vaccination, efficacy, safety, health problem

## Abstract

With the progressive lengthening of the average age of the population, especially in some countries such as Italy, vaccination of the elderly is a fixed point on which most of the public health efforts are concentrating as epidemic infectious diseases, especially those of the winter, have a major impact on the progression of severe disease, hospitalization, and death. The protection of the elderly against acute infectious diseases should not only limit mortality but also have a positive impact on the fragility of these people in terms of less disability and fewer care needs. However, vaccination of the elderly population differs in efficacy and safety compared to that of other population categories since aging and the consequent loss of efficiency of the immune system lead to a reduction in the immunogenicity of vaccines without achieving a lasting antibody coverage. There are various strategies to avoid the failure of immunization by vaccines such as resorting to supplementary doses with adjuvant vaccines, increasing the dosage of the antigen used, or choosing to inoculate the serum relying on various routes of administration of the vaccine. Vaccination in the elderly is also an important factor in light of growing antibiotic resistance because it can indirectly contribute to combating antibiotic resistance, reducing theoretically the use of those agents. Furthermore, vaccination in old age reduces mortality from infectious diseases preventable with vaccines and reduces the same rate of resistance to antibiotics. Given the importance and complexity of the topic, in this review, we will deal with the main aspects of vaccination in the elderly and how it can influence mortality and healthcare costs, especially in those countries where population aging is more evident. Therefore, we conducted a systematic literature search in PubMed to identify all types of studies published up to 31 May 2023 that examined the association between vaccination and the elderly. Data extraction and quality assessment were conducted by two reviewers (PC and TC) who independently extracted the following data and assessed the quality of each study.

## 1. Introduction

Vaccination in the elderly is an important public health aspect for the protection of this category of individuals from various infectious diseases, including those caused by viruses such as influenza, pneumonia, and, more recently, COVID-19. Vaccines can significantly reduce the risk of serious illness, hospitalization, and death in older individuals [1,2] (Figure 1). Especially in countries at greater risk of population aging, vaccination is recommended against pathogens which more frequently, based on their circulation, risk exposing frail people to often serious pathological conditions. However, vaccination of the elderly poses several problems regarding the safety and efficacy of the main serums used. In fact, it is known that in the marketing phase, each vaccine is mainly used on young subjects with an efficient immune system and therefore capable of developing high antibody titers [1,2]. Instead, as regards the efficacy of vaccines in the elderly population, there are no sufficiently valid data demonstrating that the quantity of antibodies produced is sufficient to confer protection from seasonal epidemic diseases (Figure 2). This phenomenon is related to the poor efficiency of the elderly subjects’ immune systems, characterized by a reduction in the production of antibodies by B lymphocytes and by poor reactivity of the cells that present the antigen such as CD4 T-helper lymphocytes, Langerhans cells, and dendritic cells [3,4,5]. The efficacy of a vaccine in the elderly population, therefore, depends on lower reactivity and number of antigen-presenting cells (APC), in particular of dendritic cells which tend to decrease with advancing age [6] but also show a reduction in functionality and consequently, a reduced production of interferon is observed [7]. The activation of T lymphocytes, in particular of CD8+ T lymphocytes, is also affected by the change linked to immunosenescence both in terms of a reduction in the absolute cell population and in a reduction in the various subsets with consequent functional alterations [8] such as a reduction in the production of cytokines [9,10,11,12,13]. B lymphocytes also undergo changes as a function of age which can be summarized in a reduced diversity and function of B cells and in the production of immunoglobulins during the activation of the humoral immune response [14,15,16]. In addition to the fundamental actors of the humoral immune response, the regulatory mechanisms of the immune response are also affected by immunosenescence. In particular, CD4+ lymphocytes (Tregs), which are the main elements of the correct immune distinction between self and non-self, may have a suboptimal response in the regulatory feedback of T lymphocyte activation and proliferation in cytokine production in the elderly [17,18] and this could lead to a decrease in the humoral response to vaccines in the elderly [19,20].

## 2. Methods

We conducted a systematic literature search in PubMed to identify all types of studies published up to 31 May 20223 that examined the association between vaccines and the elderly. We used the following search: (‘Vaccines ‘ OR ‘Vaccination coverage’ OR ‘Immunization’) AND (‘elderly’ OR ‘geriatric patients’ OR ‘frail patients’ OR ‘frailty’). The inclusion criteria for the evaluation of each single contribution were the following: clinical, molecular, experimental, observational studies and meta-analyses. The exclusion criteria were as follows: comments, animal studies, or case reports. Studies that are duplicated or repeated or have great similarity in sample or content to another study. Data extraction and quality assessment were conducted by two reviewers (PC and TC) who independently extracted the following data and assessed the quality of each study. Any discrepancies between reviewers in research selection, quality assessment, or data extraction were addressed by re-evaluating the original with two other authors (GM and OP). The supervision of all the work carried out was carried out by MG, the creator, and the coordinator of the research.

## 3. Efficacy of Vaccines in the Elderly

Immunosenescence is not the only factor that can influence the efficacy of vaccines in the elderly but the latter depends both on the vaccine used and on some individual factors such as age, underlying health conditions, immune status, and the specific characteristics of the whey [21,22,23]. However, vaccines have been shown to be effective in reducing the risk of infections and related complications in the elderly [5,23,24,25,26,27]. The flu vaccination is undoubtedly that which has been studied and known for the longest time; it is known that when carried out annually, it is able to reduce complications, hospitalizations, and deaths due to this infection [28,29]. Although vaccine effectiveness may be lower in older adults than younger individuals due to age-related changes in the immune system, getting vaccinated can still provide significant benefits, reducing the risk of serious disease in less compromised older adults and minimizing the annual risk of frailty [30,31]. A meta-analysis showed that the flu vaccine was capable of inducing effective immunization in 51% of individuals up to 65 years of age and only 43% of people over 65 years of age [32]. In addition to the annual influenza vaccination, it is important to associate the single-dose pneumococcal vaccination, which in the elderly may be sufficient to provide persistent immunization [33,34]. Pneumococcal vaccines, including PCV13 and PPSV23, have been shown to reduce the risk of pneumococcal infections in the elderly while also preventing secondary manifestations of pneumonia such as meningitis and bloodstream infections which can be serious and life-threatening [35]. Approximately 4500 of 24,000 cases of pneumococcal infection disease (IPD) die in the United States each year among adults over the age of 50 in the United States results [36]. These data are sufficient to support the delivery of pneumococcal vaccines to the elderly and adults with chronic illnesses. The largest case-control study [37] of vaccine efficacy showed that there is a 56% reduction in the risk of pneumonia. Other studies [38,39] suggest that decreased rates of pneumonia and IPD among young children after the introduction of PCV7 have reduced the risk of IPD among the elderly and protein conjugate vaccines have provided substantial direct benefits to children and indirect benefits for adults. Of extreme importance for the elderly population is vaccination against Herpes zoster, which is able to reduce the cases of post-herpetic neuralgia by about 65%, i.e., one of the most frequent and debilitating complications of the disease and by about 50% of all clinical cases of shingles [40]. As with previous age groups, there may be particular risk conditions that indicate a recommendation for meningococcal, hepatitis A, or hepatitis B vaccinations [41]. Additionally, COVID-19 vaccines, such as those developed by Pfizer-BioNTech, Moderna, and Johnson & Johnson, have also shown high efficacy rates in older adults in reducing the risk of serious illness, hospitalization, and death from COVID-19, with a 37–43% (33% to 62%) reduction in the risk of death in those who had been vaccinated. Participants who received one dose of the AstraZeneca vaccine had an additional 37% reduction (from 3% to 59%) in the risk of emergency hospitalization for severe disease and all complications [42,43].

## 4. Safety of Vaccines in the Elderly

Before vaccines are approved for use, they undergo rigorous testing in clinical trials that include participants of various age groups, including the elderly [44,45,46]. There are also some vaccine monitoring systems that are implemented once such sera have been licensed and introduced into public vaccination programs [47]. In addition, systems exist to evaluate vaccine safety that includes post-marketing surveillance, reporting of potential adverse events following vaccination, and, in the post-administration period, helping to implement ongoing knowledge about vaccine safety in the elderly population [48,49]. This continuous monitoring has shown that the side effects of vaccines are mostly mild and temporary and limited to the first hours of immunization [50,51]. The most recorded effects include pain or redness and swelling at the injection site, fatigue, headache, body aches, and fever [52]. These side effects usually resolve on their own within a few days and are signs that can be interpreted as activation of the immunization system. The serious adverse events recorded are quite rare and concern subjects already predisposed to vaccine reactions, the result of previous reactions [53]. These subjects are implicitly excluded a priori from vaccine treatments by evaluating more appropriate alternative therapeutic actions [54].

## 5. Mortality and Vaccines in the Elderly

Vaccines play a crucial role in reducing mortality rates in the elderly by preventing serious infections and associated complications. For this reason, the data available in the literature on the use of the main vaccines also confirm the positive impact of this therapeutic strategy in reducing annual mortality in the population due to preventable diseases and complications or conditions associated with them [55,56,57,58]. The flu vaccine has always been able to protect against serious evolutionary conditions that originate from viral contagion and at the same time reduce hospitalization and frailty rates, especially in the elderly (3.8 influenza-associated excess C and R hospitalizations for each hospitalization coded with an influenza-specific diagnosis in patients aged ≥65 years) [54,55]. Although vaccination immunization in elderly patients does not reach optimal values, it still means having effective protection in reducing the severity of the disease and the risk of complications that can lead to mortality [59,60]. Also, for the anti-pneumococcal vaccination, a reduction in all the morbid conditions related to the pneumococcus was observed over time, including even those potentially more fatal and those with fulminant onset, contributing to a reduction in the risk of mortality in the elderly [58,61,62]. Vaccination against herpes zoster indirectly reduces the risk of associated mortality in the elderly [63,64,65,66]. On the other hand, it is known that the elderly population has paid a high price in terms of mortality, in terms of persistence of serious sequelae deriving from the COVID-19 pandemic, especially in the pre-vaccination era [67,68,69,70,71,72,73].In particular, variables commonly reported for adverse COVID-19 outcomes or increased frailty included age >75 (OR: 2.65, 95% CI: 1.81–3.90), male sex (OR: 2.05, 95% CI: 1.39–3.04), and severe obesity (OR: 2.57, 95% CI: 1.31–5.05). Active cancer (OR: 1.46, 95% CI: 1.04–2.04) was associated with an increased risk of serious outcomes. A number of common symptoms and vital measures (respiratory rate and SpO2) also suggested elevated risk profiles [74]. A significant mortality improvement at the time of the introduction of the first vaccines was primarily registered for the most fragile elderly and more exposed to the risk of serious and complicated forms related to this infection [74,75,76,77]. COVID-19 vaccines, such as those developed by Pfizer-BioNTech, Moderna, and Johnson & Johnson, have shown high efficacy in reducing the risk of serious illness, hospitalization, and death [78,79,80]. In particular, in fully vaccinated elderly and frail residents there was an estimated 88.4% (95% CI: 74.9–94.7%) reduction in hospitalizations and a 97.0% (95% CI: 91 0.7–98.9%) in deaths [80,81].

## 6. Vaccination and Antibiotics Resistance

Antibiotic resistance and vaccination appear to be two separate topics but inevitably are two interrelated issues when it comes to tackling infectious disease in the elderly, particularly in this post-pandemic period where both frailty and resistance to antibiotics are unresolved weaknesses in our healthcare systems [82,83,84]. The higher rate of preventable infectious disease, the higher the consumption of various drugs including antibiotics which are sometimes used improperly and intentionally, thus increasing their resistance [85]. Currently, more and more bacteria and fungi are developing biological mechanisms to resist the effects of antibiotics, rendering them ineffective in treating infections [85,86,87]. This issue is of global public health concern and has implications for all age groups, including the elderly who have traditionally been more susceptible to antibiotic-resistant infections due to factors such as a weakened immune system, higher rates of nosocomial infections, and greater fragility [88]. From what has been said, it is clear that vaccination of the elderly population can indirectly contribute to combating resistance to antibiotics, reducing and rationalizing their consumption [89,90]. Furthermore, also from a biological point of view, by stimulating the immune system in recognizing the antigens of the main seasonal pathogens, vaccines make the immune cell less vulnerable and less passive in recognizing the various pathogens, reducing the probability of global infection and therefore the consequent need for antibiotics [91,92]. Furthermore, there are many infections that can be prevented with vaccines such as pneumonia, meningitis, and some respiratory infections as well as various forms of septicemia [93,94,95]. By vaccinating the elderly and at-risk sections of the population against these diseases, the incidence of infections can be reduced, thus potentially decreasing the need for antibiotics and reducing the selection pressure for antibiotic-resistant bacteria [96,97]. The results of a recent meta-analysis show that the effect of influenza vaccination on the number of antibiotic prescriptions or days of antibiotic use (Ratio Means (RoM) 0.71, 95% CI 0.62–0.83) is stronger compared to the effect of pneumococcal vaccination (RoM 0.92, 95% CI 0.85–1.00). These studies also confirm a reduction in the percentage of people receiving antibiotics after influenza vaccination (hazard ratio (RR) 0.63, 95% CI 0.51–0.79). The effect of influenza vaccination in the European and American regions ranged from RoM 0.63 and 0.87 to RR 0.70 and 0.66, respectively [98]. As we have already observed, the excessive consumption of antibiotics is often linked to incongruous prescriptions both by etiology (the symptoms are often largely due to viral infections), molecule (the choice of the active ingredient is often not targeted and its non-quantifiable therapeutic effect), the dosage used (most antibiotics are not used in therapeutic doses), and finally by administration times, which are not always compliant with existing guidelines [84,99]. Most of the incongruous prescribing is related to the fear of preventing more serious complications. Prevention efforts, instead of being directed at promoting timely vaccination, are directed at associating the use of antibiotics with other symptomatic therapies from the first appearance of symptoms [100,101]. In this sense, vaccination must be seen as the first safeguarding option for the prevention of serious disease forms with repercussions not only in terms of the request for antibiotic treatment but also in terms of contributing to the overall efforts in the fight against antibiotic resistance [84].

## 7. Vaccination in the Immunocompromised Elderly

Vaccination in the immunocompromised elderly may provide important safety and benefits, although there are some considerations to keep in mind [100,101,102,103,104]. In terms of security, vaccines are now subjected to safety tests before being approved for use in individuals with a primary or secondary immunocompromised state such as iatrogenic [105]. In some cases, some vaccines may not be recommended due to potential risks associated with the individual’s immune system response such as during the administration of the influenza vaccine (flu vaccine); measles, mumps, and rubella (MMR) vaccine; tetanus, diphtheria, and pertussis vaccine (Tdap); and finally, COVID-19 vaccines (Pfizer-BioNTech, Moderna, Johnson & Johnson, etc.) as well as the yellow fever vaccine. On the other hand, vaccination in the immunocompromised elderly offers several advantages in terms of protection against preventable diseases and the serious complications associated with them [104,105]. An effective vaccination campaign then contributes to herd immunity, creating a protective immunogenic barrier around the fragile and immunocompromised subject which limits the spread of infectious diseases, both in the community and above all in hospital structures for acute cases in residences for the elderly where the presence of immunosuppressed individuals is preponderant [102,103]. Vaccinating such individuals also helps to ease the burden of healthcare costs associated with more complex healthcare needs [106,107]. It was observed that adherence to vaccination induces an improvement in the quality of life in the groups of immunocompromised elderly people and their families and this translates into greater independence of these subjects and greater autonomy and mobility which limits physical inactivity and frailty in individuals [108,109].

## 8. Vaccination Coverage: A Public Health Problem

Up to now, we have said that vaccinations provide positive effects in terms of significant gains in the effectiveness of invested resources and in terms of the benefit/risk ratio. However, in public opinion, the perception of the real usefulness of vaccines does not always correspond to what the scientific community has been able to observe in decades of vaccination campaigns [47,110]. There are several factors that can influence vaccination coverage [111]. These include the availability of vaccines, access to health services, public confidence in vaccines, the dissemination of misinformation or anti-vaccine information, and the vaccination policies adopted by health authorities [112]. Studies indicate that there is still a need to expand vaccination coverage in a population that is still not convinced of the real usefulness of this public health device [113,114,115]. Vaccination coverage is indeed a very important public health issue. It refers to the percentage of a given population that has received a specific vaccination. Adequate vaccination coverage is essential to prevent the spread of infectious diseases and protect the health of the community as a whole [113,114,115,116]. When vaccination coverage is high, herd immunity is achieved, which is a form of indirect protection for those who cannot be vaccinated due to legitimate reasons such as being too young age or having health problems. Herd immunity occurs when a sufficient percentage of the population is immunized, making it more difficult for a pathogen to infect and spread within the community [115]. However, when vaccination coverage is low, disease outbreaks can occur that could otherwise be prevented. This puts not only unvaccinated people at risk but also those who cannot be vaccinated for valid reasons. To overcome the problem of low adherence to vaccination protocols, it is important to adopt a multi-layered approach and involve different stakeholders, including local governments, health authorities, health professionals, organizations for the elderly, and families, to promote high vaccination coverage among the elderly [116]. In some health systems, there are a number of vaccine implementation strategies that consist of educational efforts for clear, accessible, and science-based information on the positive effects and safety of vaccines for older adults, using information brochures, websites, advertisements, public meetings, and television programs. To this list, information from family doctors, nurses, and health professionals duly updated on vaccination guidelines should be added as well as the possibility of offering vaccines during routine medical visits [117,118,119]. In fact, one of the important problems is to make vaccines more easily accessible to the elderly by organizing vaccination sessions at health facilities, nursing homes, community centers for the elderly, or other places frequented by the elderly. In addition, consideration should be given to arranging home immunization services for those with mobility difficulties [117,118,119]. It would be helpful to conduct awareness campaigns targeting older people and involve family members and carers that older people often rely on. Additionally, access to vaccination should be facilitated for caregivers who may be younger and at high risk of transmitting the disease to older adults [119]. For each vaccination campaign, it is important to preliminarily evaluate the cost-effectiveness ratio. Cost-effectiveness analysis involves comparing the costs and health benefits of vaccination with alternative interventions or the absence of a vaccination program [120,121,122,123]. This report represents a critical consideration for public health policy. Vaccination programs aim to prevent the spread of infectious diseases and avoid bearing related health and economic burdens [120,121,122,123]. Evaluating the cost-effectiveness of these programs involves evaluating the costs incurred and the health benefits gained from vaccination, usually measured in terms of the quality-adjusted life years (QALYs) gained [120,121,122,123,124]. Several factors contribute to the cost-effectiveness of vaccination programs:Burden of disease: The impact of a disease on morbidity, mortality, and healthcare costs influences the potential benefits of vaccination;Vaccine effectiveness: The effectiveness of a vaccine in preventing disease transmission and reducing its severity directly affects its cost-effectiveness;Immunization coverage: Higher vaccination rates within a population lead to greater overall protection and more cost-effective outcomes;Vaccine price: The cost of the vaccine itself can have a significant impact on the cost-effectiveness of a vaccination program.Healthcare costs: Vaccinations reduce the need for medical care, hospital stays, and related costs associated with treating vaccine-preventable diseases;Lost productivity: Vaccination prevents lost working days and productivity losses due to illness which has economic implications;Herd immunity: When a sufficient percentage of a population is immunized, herd immunity can develop, providing indirect protection to those who are not immunized;Vaccine safety: Addressing vaccine safety concerns is critical to maintaining public trust and encouraging participation.

On the other hand, low vaccination coverage can have significant consequences, leading to disease outbreaks and associated health, economic, and social impacts. Potential consequences of this phenomenon are:The resurgence of vaccine-preventable diseases: Low vaccination coverage can result in the resurgence of vaccine-preventable diseases (VPDs) such as measles, mumps, pertussis, and polio. These diseases can spread rapidly in communities with low immunity levels [125];Increased Disease Transmission: When vaccination rates are low, the overall level of community immunity (herd immunity) decreases. This allows diseases to spread more easily, including to vulnerable populations such as infants, elderly individuals, and those with compromised immune systems [126];Outbreak-Related Healthcare Costs: Disease outbreaks can strain healthcare systems, leading to increased hospitalizations and medical costs. Low vaccination coverage can exacerbate these costs, burdening both individuals and healthcare facilities [127];Impact on Vulnerable Populations: Low vaccination coverage disproportionately affects vulnerable populations, including infants who are too young to be vaccinated and those with medical contraindications to vaccination. These individuals are at higher risk of severe disease and complications [128];Loss of Public Trust: Ongoing vaccine hesitancy and low vaccination rates erode public trust in vaccines and public health systems. This can further reduce vaccine uptake, creating a vicious cycle of declining coverage and increased disease risk [129];International Disease Spread: Low vaccination coverage in one region or country can lead to international disease spread. Travelers can carry diseases across borders, resulting in outbreaks in areas with higher vaccination coverage [130];Impact on Eradication Efforts: For diseases targeted for eradication, such as polio and measles, low vaccination coverage hinders progress. It can make it difficult to achieve and sustain disease elimination goals [131].

In summary, low vaccination coverage can lead to the reemergence of vaccine-preventable diseases, increased disease transmission, healthcare costs, and a range of negative consequences for public health. These consequences underscore the importance of achieving and maintaining high vaccination coverage rates to protect both individuals and communities.

## 9. Take Home Messages

Vaccination is an important preventive measure to protect the health and well-being of older adults. It not only reduces the risk of severe infections but also decreases mortality rates associated with vaccine-preventable diseases;Many vaccine-preventable infections, such as pneumonia, meningitis, and certain respiratory and bloodstream infections, are commonly associated with antibiotic use. By vaccinating older adults against these diseases, the incidence of infections can be reduced, thereby potentially decreasing the need for antibiotics and reducing the selection pressure for antibiotic-resistant bacteria;The safety of vaccines in the elderly has been extensively studied and vaccines are generally considered safe for older adults. Vaccination plays a crucial role in protecting older individuals from vaccine-preventable diseases and their associated complications;Vaccines can significantly reduce the risk of infections and related complications. The efficacy of vaccines can be influenced by factors such as age, underlying health conditions, immune status, and the specific vaccine’s characteristics;Promoting the integration of various institutions and professionals in the field of health care should be considered to maintain successful national immunization programs;Promote a massive information campaign on the part of the scientific world, state health authorities, and various health operators which makes vaccination perceived as a healthy element of life, using high-quality forms of vaccination advice and thus further contributing to the increase in immunization rates in the elderly population and their caregivers.

## 10. Conclusions

Vaccines play a crucial role in reducing mortality rates in the elderly by preventing severe infections and associated complications. Any vaccine-preventable infections, such as pneumonia, meningitis, and certain respiratory and bloodstream infections, are commonly associated with antibiotic use. By vaccinating older adults against these diseases, the incidence of infections can be reduced, thereby potentially decreasing the need for antibiotics and reducing the selection pressure for antibiotic-resistant bacteria. Older adults may be more susceptible to antibiotic-resistant infections due to factors such as weakened immune systems, higher rates of healthcare-associated infections, and more frequent antibiotic use. Despite these positive effects, vaccine resistance is observed specifically in the elderly population. Age-related changes in the immune system, the individual’s immune response, and the individual’s overall health status often limit vaccine efficacy. Certain medical conditions, such as immunodeficiency or chronic diseases, may impair the immune system’s ability to mount a robust response to vaccines. As a result, the level of protection provided by vaccines may be reduced in these individuals. Despite these factors, vaccination remains crucial for older adults as it can still provide significant benefits in terms of reducing the risk of severe illnesses, hospitalizations, and complications.

## Figures and Tables

**Figure 1 vaccines-11-01412-f001:**
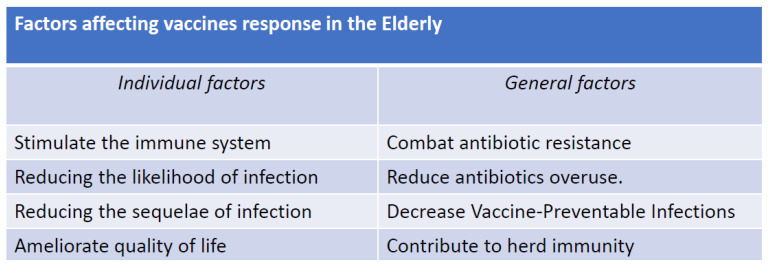
Importance of vaccination in the elderly.

**Figure 2 vaccines-11-01412-f002:**
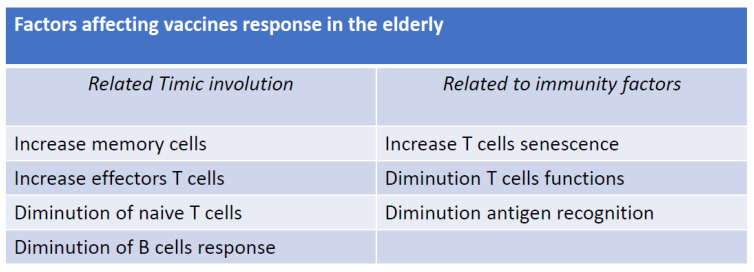
Resistance to vaccine response in the elderly.

## Data Availability

Not applicable.

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
