# Peer review of "Efficacy and Safety of Vaccinations in Geriatric Patients: A Literature Review"

_vaccines, 2023, doi:10.3390/vaccines11091412_

Round 1

Reviewer 1 Report

The title of the review article "Vaccination Coverage: A Public Health Problem in the Elderly Population?" seems intriguing, as it addresses a significant public health concern regarding vaccination coverage in the elderly population. However, one notable shortcoming of the title is its lack of specificity. It fails to provide essential details about the review's scope, context, or focus. The term "vaccination coverage" remains ambiguous, and the title does not indicate which specific vaccinations or regions are being referred to. Additionally, it does not clarify the time frame or population demographics being considered.

The abstract of the review article provides a comprehensive overview of the significance of vaccination in the elderly population, especially in countries with aging demographics. It highlights the importance of protecting the elderly against infectious diseases to reduce mortality and healthcare costs. Furthermore, the abstract lacks a clear statement of the review's methodology, which would strengthen the credibility of the findings.

One crucial aspect that needs to be addressed in the review article is the lack of a comprehensive description regarding the methodology employed for data search and selection. The manuscript fails to mention the keywords used during the literature search, the databases consulted for data exploration, and the inclusion and exclusion criteria applied to identify relevant studies. Such omissions impede the transparency and replicability of the review process, as readers and researchers need clarity on how the evidence was gathered and which sources were considered. Without a clear outline of the methodological approach, it becomes challenging for the audience to assess the rigor of the review and the reliability of the findings presented. Therefore, it is recommended that the authors provide a detailed account of their data search strategy, including the databases utilized, the search terms employed, and the criteria used to select studies for inclusion in the review.

The introduction section of the review article provides a comprehensive overview of the importance of vaccination in the elderly and the challenges associated with immunization in this population. The author highlights the significance of protecting older adults from infectious diseases, particularly in countries with aging populations. The discussion on the limitations of current vaccines in the elderly due to the decline in immune function and reduced efficacy is well-presented, supported by references to relevant studies. However, the significant major aspect of this section is the lack of clarity in organization and structure. The ideas seem scattered and could benefit from a more precise flow of information. The introduction would be more robust if the author provided more direct evidence and data to support their claims.

The heading "Efficacy of Vaccines in the Elderly" starts with repetition and lacks a clear transition from the previous section. The author mentions wide variability in clinical response and effectiveness but does not provide concrete data or studies to support this claim. Furthermore, the text mentions individual factors influencing vaccine efficacy without specifying what those factors are. The paragraph could benefit from more specific examples and evidence from studies that demonstrate varying vaccine efficacy in older people due to immunosenescence and individual characteristics.

Similarly, the heading "Safety of Vaccines in the Elderly" begins with unnecessary repetition, indicating a lack of seamless flow between the sections. The paragraph on vaccine safety in the elderly is relatively short and provides only general information. The author mentions that vaccines are generally considered safe for older adults but fails to elaborate on specific safety concerns that may arise in this population due to age-related factors. Moreover, the paragraph lacks direct references to studies or data supporting the claim that adverse events in the elderly are mostly mild and temporary. Providing specific evidence would enhance the credibility of the safety assessment.

The heading "Mortality and Vaccine in the Elderly" presents relevant information on the impact of vaccines on reducing mortality rates in older adults.

The heading "Vaccination and Antibiotics Resistance" discusses an important and relevant connection between vaccination and antibiotic resistance. However, the paragraph could benefit from a clearer structure and a more coherent presentation of the information. The author should introduce the link between vaccines and the reduction of infectious diseases, leading to decreased antibiotic usage and, consequently, less antibiotic resistance. Additionally, the paragraph would benefit from specific examples and evidence supporting the claim that vaccination can reduce the need for antibiotics in the elderly population. Including concrete data on the reduction in antibiotic prescriptions and the incidence of antibiotic-resistant infections post-vaccination would enhance the argument's credibility.

The heading "Vaccination Coverage: A Public Health Problem" provides a clear focus on vaccination coverage as a public health concern. However, while the text mentions the importance of herd immunity, it does not delve into the potential consequences of low vaccination coverage regarding disease outbreaks and the health risks faced by both vaccinated and unvaccinated individuals. Including this information would add depth to the discussion and highlight the urgent need to address the issue of low adherence to vaccination protocols in the elderly population.

The conclusion of the review article provides a concise summary of the crucial role of vaccines in reducing mortality rates in the elderly by preventing severe infections and associated complications. The paragraph emphasizes the positive effects of vaccination in reducing the need for antibiotics and the selection pressure for antibiotic-resistant bacteria.

I have observed that the bibliography in the reviewed article does not adhere to the journal's formatting guidelines. I would recommend that the authors carefully review and revise the bibliography to ensure that each reference follows the correct citation style prescribed by the journal.

Requires minor editing.

Author Response

Dear Editor and Dear Authors,

Thank you very much for your crucial revision.

We report, now the response for Reviewer’s revision.

Thank you

REVIEWER 1.

the title of the review article "Vaccination Coverage: A Public Health Problem in the Elderly Population?" seems intriguing, as it addresses a significant public health concern regarding vaccination coverage in the elderly population. However, one notable shortcoming of the title is its lack of specificity. It fails to provide essential details about the review's scope, context, or focus. The term "vaccination coverage" remains ambiguous, and the title does not indicate which specific vaccinations or regions are being referred to. Additionally, it does not clarify the time frame or population demographics being considered.

R:  In accordance with the reviewer's suggestions, the title has been changed.

The abstract of the review article provides a comprehensive overview of the significance of vaccination in the elderly population, especially in countries with aging demographics. It highlights the importance of protecting the elderly against infectious diseases to reduce mortality and healthcare costs. Furthermore, the abstract lacks a clear statement of the review's methodology, which would strengthen the credibility of the findings.

R: In accordance with the reviewer's suggestions, the methodology used for the composition of the paper was explained in the abstract.

One crucial aspect that needs to be addressed in the review article is the lack of a comprehensive description regarding the methodology employed for data search and selection. The manuscript fails to mention the keywords used during the literature search, the databases consulted for data exploration, and the inclusion and exclusion criteria applied to identify relevant studies. Such omissions impede the transparency and replicability of the review process, as readers and researchers need clarity on how the evidence was gathered and which sources were considered. Without a clear outline of the methodological approach, it becomes challenging for the audience to assess the rigor of the review and the reliability of the findings presented. Therefore, it is recommended that the authors provide a detailed account of their data search strategy, including the databases utilized, the search terms employed, and the criteria used to select studies for inclusion in the review.

R: In accordance with the reviewer's suggestions, paragraph 2 dedicated to the research methods used in the study was created

The introduction section of the review article provides a comprehensive overview of the importance of vaccination in the elderly and the challenges associated with immunization in this population. The author highlights the significance of protecting older adults from infectious diseases, particularly in countries with aging populations. The discussion on the limitations of current vaccines in the elderly due to the decline in immune function and reduced efficacy is well-presented, supported by references to relevant studies. However, the significant major aspect of this section is the lack of clarity in organization and structure. The ideas seem scattered and could benefit from a more precise flow of information. The introduction would be more robust if the author provided more direct evidence and data to support their claims.

R: In accordance with the reviewer's suggestions, the introductory chapter has been revised in its structure

The heading "Efficacy of Vaccines in the Elderly" starts with repetition and lacks a clear transition from the previous section. The author mentions wide variability in clinical response and effectiveness but does not provide concrete data or studies to support this claim. Furthermore, the text mentions individual factors influencing vaccine efficacy without specifying what those factors are. The paragraph could benefit from more specific examples and evidence from studies that demonstrate varying vaccine efficacy in older people due to immunosenescence and individual characteristics.

R: In accordance with the reviewer's suggestions, the "Efficacy of Vaccines in the Elderly" chapter has been revised in its structure.

Similarly, the heading "Safety of Vaccines in the Elderly" begins with unnecessary repetition, indicating a lack of seamless flow between the sections. The paragraph on vaccine safety in the elderly is relatively short and provides only general information. The author mentions that vaccines are generally considered safe for older adults but fails to elaborate on specific safety concerns that may arise in this population due to age-related factors. Moreover, the paragraph lacks direct references to studies or data supporting the claim that adverse events in the elderly are mostly mild and temporary. Providing specific evidence would enhance the credibility of the safety assessment.

R: In accordance with the reviewer's suggestions, the " Safety of Vaccines in the Elderly " chapter has been revised in its structure.

The heading "Mortality and Vaccine in the Elderly" presents relevant information on the impact of vaccines on reducing mortality rates in older adults.

R: No comments

The heading "Vaccination and Antibiotics Resistance" discusses an important and relevant connection between vaccination and antibiotic resistance. However, the paragraph could benefit from a clearer structure and a more coherent presentation of the information. The author should introduce the link between vaccines and the reduction of infectious diseases, leading to decreased antibiotic usage and, consequently, less antibiotic resistance. Additionally, the paragraph would benefit from specific examples and evidence supporting the claim that vaccination can reduce the need for antibiotics in the elderly population. Including concrete data on the reduction in antibiotic prescriptions and the incidence of antibiotic-resistant infections, post-vaccination would enhance the argument's credibility.

R: In accordance with the reviewer's suggestions, the " Vaccination and antibiotic " chapter has been revised in its structure and more details on antibiotic consumption reduction have been added.

The heading "Vaccination Coverage: A Public Health Problem" provides a clear focus on vaccination coverage as a public health concern. However, while the text mentions the importance of herd immunity, it does not delve into the potential consequences of low vaccination coverage regarding disease outbreaks and the health risks faced by both vaccinated and unvaccinated individuals. Including this information would add depth to the discussion and highlight the urgent need to address the issue of low adherence to vaccination protocols in the elderly population.

R: In accordance with the reviewer's suggestions, the " Vaccination Coverage: A Public Health Problem" chapter has been revised in its structure and more details on disease outbreaks caused by low vaccination coverage, have been added.

The conclusion of the review article provides a concise summary of the crucial role of vaccines in reducing mortality rates in the elderly by preventing severe infections and associated complications. The paragraph emphasizes the positive effects of vaccination in reducing the need for antibiotics and the selection pressure for antibiotic-resistant bacteria.

I have observed that the bibliography in the reviewed article does not adhere to the journal's formatting guidelines. I would recommend that the authors carefully review and revise the bibliography to ensure that each reference follows the correct citation style prescribed by the journal.

R: In accordance with the reviewer's suggestions, the bibliography will be revised during the proofreading phase, in case of acceptance of the paper.

Many thanks

Tiziana Ciarambino

MD, PhD

Reviewer 2 Report

Dear Editor,

Dear Authors,

 Thank You for the opportunity to review a manuscript entitled ‘Vaccination Coverage: A Public Health Problem in the Elderly Population?’

The manuscript is well written and in general fits the quality criteria to be published in Vaccines, there are some issues, however, which I suggest taking into account to improve its scientific soundness:

*-the title: it states ‘vaccination coverage’ but this issue is narrowly discussed in the manuscript . There are two options for consideration: 1) in a paragraph 7 with the description of the coverage levels across different populations/regions in the elderly populations and the factors associated with the coverage, or 2) rename the title adding ‘vaccination efficacy and safety’ as these are the main issues being in the focus of the manuscript;

*-the title: add a term ‘literature review’ to inform the reader what is the general intent of the manuscript;

*-page2 lines65-68: … all these steps are compromised in 65 the elderly patient because most of the APC cells and in particular of the DC are reduced in number and this reduction is progressive with advancing age – reference required; and ‘these cells are also not functioning well since it has been observed that they have a very low production of interferon’ – reference required.;

*-p3-l.100-101: some additional information about how the effectiveness of vaccines changes with ageing would be useful;

*-p3-l.112-113: provide, please, the amount of risk reduction for severe illness and/or for risk of frailty. The same topic is repeated on page 4-line 164 without data.

*-p4-l.130: it would be useful to know what is the approximate amount of reducing the risk of hospitalization by the vaccination

*p4-l.152-152: provide please the frequency of serious adverse events … there are two imprecise statements there as ‘… are quite rare …’ followed by ‘… they are extremely rare …’;

*p6-l.221-222: ‘In some cases, some vaccines may not be recommended due to potential risks associated with the individual's immune system response’ – add, please, some examples;

*paragraph 7 Vaccination Coverage: considering the expected/observed benefits from vaccination some considerations and comments on cost-effectiveness of vaccination programs would be worth mentioning;

*p7-l.276: second sentence: any->Many ? … typing error ?

*p7-l.307-308: ‘Vaccines can significantly reduce the risk of infections and related complications, they may not provide 100% protection for every individual’ – the protection levels are not provided and discussed. I suggest deleting this sentence … or add a paragraph with different protection levels (across different vaccination types/doses/administration schemes) and show the ranges here.

*p7-l.314: ‘the active initiation of the discussion on vaccines’ … it is not about the initiation of the discussion, it is rather about popularity and perceiving being vaccinated as an element of healthy lifestyle. I suggest rewriting this take home message.

Overall it is an interesting paper.

Reviewer

There are some repetitions which might be avoided on page 2 lines 58 & 60 'therefore'.

*p7-l.276: second sentence: any->Many ? … typing error ?

Author Response

Dear Editor and Dear Authors,

Thank you very much for your crucial revision.

We report, now the response for Reviewer’s revision.

Thank you

REVIEWER 2

The manuscript is well written and in general, fits the quality criteria to be published in Vaccines, there are some issues, however, which I suggest taking into account to improve its scientific soundness:

*-the title: it states ‘vaccination coverage’ but this issue is narrowly discussed in the manuscript. There are two options for consideration: 1) in paragraph 7 with the description of the coverage levels across different populations/regions in the elderly populations and the factors associated with the coverage, or 2) rename the title adding ‘vaccination efficacy and safety as these are the main issues being in the focus of the manuscript;

R:  In accordance with the reviewer's suggestions, the title has been changed.

*-the title: add a term ‘literature review’ to inform the reader what is the general intent of the manuscript;

R:  In accordance with the reviewer's suggestions, the title has been changed.

*-page2 lines65-68: … all these steps are compromised in 65 the elderly patient because most of the APC cells and in particular of the DC are reduced in number and this reduction is progressive with advancing age – reference required; and ‘these cells are also not functioning well since it has been observed that they have a very low production of interferon’ – reference required.;

R: In accordance with the reviewer's suggestions, two references have been added to justify the statement.

*-p3-l.100-101: some additional information about how the effectiveness of vaccines changes with aging would be useful;

R: In accordance with the reviewer's suggestions, some additional information on how vaccine efficacy changes with aging has been added;

*-p3-l.112-113: provide, please, the amount of risk reduction for severe illness and/or for risk of frailty. The same topic is repeated on page 4-line 164 without data.

R: In accordance with the reviewer's suggestions, some additional information about risk reduction for severe illness and/or for risk of frailty has been added;

*-p4-l.130: it would be useful to know what is the approximate amount of reducing the risk of hospitalization by the vaccination

R: In accordance with the reviewer's suggestions, some additional information about reducing the risk of hospitalization by the vaccination has been added;

*p4-l.152-152: provide please the frequency of serious adverse events … there are two imprecise statements there as ‘… are quite rare …’ followed by ‘… they are extremely rare …’;

R: In accordance with the reviewer's suggestions, the sentence has been revised.

*p6-l.221-222: ‘In some cases, some vaccines may not be recommended due to potential risks associated with the individual's immune system response’ – add, please, some examples;

R: In accordance with the reviewer's suggestions, some examples of vaccination individual's immune system response have been added;

*paragraph 7 Vaccination Coverage: considering the expected/observed benefits from vaccination some considerations and comments on the cost-effectiveness of vaccination programs would be worth mentioning;

R: In accordance with the reviewer's suggestions, some additional information on the cost-effectiveness relationship of vaccine campaigns has been added;

*p7-l.307-308: ‘Vaccines can significantly reduce the risk of infections and related complications, they may not provide 100% protection for every individual’ – the protection levels are not provided and discussed. I suggest deleting this sentence … or add a paragraph with different protection levels (across different vaccination types/doses/administration schemes) and show the ranges here.

R: In accordance with the reviewer's suggestions, the sentence has been revised.

*p7-l.314: ‘the active initiation of the discussion on vaccines’ … it is not about the initiation of the discussion, it is rather about popularity and perceiving being vaccinated as an element of healthy lifestyle. I suggest rewriting this take home message.

R: In accordance with the reviewer's suggestions, the sentence has been revised.

Overall, it is an interesting paper.

Comments on the Quality of English Language

There are some repetitions which might be avoided on page 2 lines 58 & 60 'therefore'.

R: In accordance with the reviewer's suggestions, the repetition has been revised.

*p7-l.276: second sentence: any->Many ? … typing error ?

R: In accordance with the reviewer's suggestions, the sentence has been revised.

Round 2

Reviewer 1 Report

I am glad to know that the authors have addressed the recommendations and have significantly improved the manuscript. I am pleased to recommend the manuscript for publication.